# A Numerical Simulation of Moisture Reduction in Fine Soil Subgrade with Wicking Geotextiles

**DOI:** 10.3390/ma17020390

**Published:** 2024-01-12

**Authors:** Chuanyi Ma, Haojie Feng, Chuan Wang, Ning Zhang, Yiyi Liu, Jinglei Li, Xia Liu, Shasha Li, Hongguang Jiang, Yixin Li

**Affiliations:** 1Shandong Hi-Speed Group Co., Ltd., Jinan 250014, China; 18710844362@163.com (C.M.); xinyu19990627@163.com (C.W.); 15339118226@163.com (N.Z.); 2School of Qilu Transportation, Shandong University, Jinan 250002, China; 202215411@mail.sdu.edu.cn (H.F.); 202015394@mail.sdu.edu.cn (Y.L.); jinglei_0428@163.com (J.L.); lishasha0902@163.com (S.L.); 3Jining Hongxiang Highway Survey and Design Research Institute Co., Ltd., Jining 272000, China; zm18364992640@163.com; 4School of Engineering, University of Warwick, Coventry CV4 7AL, UK; yixin.li@warwick.ac.uk

**Keywords:** wicking geotextiles, numerical simulation, humidity control, fine-grained soil, matric suction

## Abstract

A new wicking geotextile is proposed to control the water content of fine-grained soil subgrade. By comparing the spatial distribution of volumetric water content and matric suction before and after the installation of the wicking geotextile, the effectiveness of the geotextile in controlling the subgrade humidity is evaluated. Firstly, the hydraulic parameters of the wicking geotextile are obtained through laboratory tests using a pressure plate apparatus. Then, a numerical model for water flow in the subgrade is established using COMSOL to obtain the spatial distribution characteristics of humidity in the subgrade under different groundwater levels (2~8 m). The results show the wicking geotextile exhibits strong hydrophilicity, low water retention, and high horizontal permeability. Compared to the subgrade without geotextile, the water content of the soil above the geotextile decreases significantly by 7.6~9.6% at groundwater levels of 4~8m, while the saturation decreases by 18.3~23.0%, and the matric suction increases by 2~2.3 times. The wicking fabric functions as an effective drainage material to serve as a capillary barrier in the cross-plane direction and an effective drainage tunnel to transport water in the in-plane direction. The dynamic resilient modulus of the subgrade increases by 23.2~43.6%. The wicking geotextile effectively absorbs and drains weakly bound water in unsaturated soil due to the matric suction difference and its horizontal drainage capacity to improve the bearing capacity of the subgrade. It suggests that using wicking geotextile for drainage and reinforcement in fine-grained soil subgrades with groundwater levels ranging from 4 to 8 m is beneficial.

## 1. Introduction

A report showed that 60% of China’s expressways require major repairs or renovations after 10~12 years of service, and 17% require such repairs after 6~8 years, indicating a shorter-than-expected service life for expressways [1]. Studies have found that subgrade moisture is one of the main causes of this problem [2,3,4]. Typically, the subgrade is compacted at its optimal moisture content to achieve optimal performance. However, during actual service, precipitation infiltration and capillary rise may cause the subgrade moisture content to reach an equilibrium humidity higher than the optimal moisture content, leading to the degradation of subgrade support performance [5,6], further resulting in a decline in the pavement’s service life. Rahman et al. [7,8] revealed that specimens compacted at +2%*w_opt_* showed a lower modulus than specimens compacted at *w_opt_*. The soil moisture condition significantly influenced the subgrade modulus and the resulting subgrade rutting, which led to the severe degradation of road service performance.

Effectively controlling the internal humidity of the subgrade is the key to ensuring its support performance. Pooni et al. [9,10] examined the durability performance of the enzymatic stabilization of expansive soils in road pavements, revealing its ability to maintain the material stiffness over moisture fluctuation subjected to moisture fluctuation. The current method of controlling the internal humidity of the subgrade is mainly through the use of a drainage layer of sand and gravel, relying on gravity potential to drain water within the road structure. However, for the vast majority of its service life, the subgrade is in an unsaturated state, and the soil skeleton produces capillary suction to water, such as the fine-grained clay commonly used in road construction, which can generate suction of over 100 kPa at a moisture content of 12%, rendering conventional drainage methods ineffective [11,12,13,14,15]. The wetting of subgrade will greatly reduce its supporting performance; Lin et al. [16] performed resilient modulus tests for a base course material containing 10% fine grains. Test results indicated that the resilient modulus value reduced from 98 MPa to 56 MPa when the water content increased from 8.9% (optimum water content 8.5%) to 10.8%. To control the internal moisture content of unsaturated subgrades, a new type of wicking geotextile has been developed, with a unique grooved cross-sectional structure [17] and hydrophilic groups that can produce capillary force, generating “core suction” to the water in the soil and actively draining it in unsaturated soil. Galinmoghadam et al. [18] used wicking geotextiles to solve the pumping and sucking problem and concluded that the drainage capacity of the wicking geotextile under unsaturated conditions was superior to traditional drainage methods. Lin et al. [19] determined the geotextile–soil–water characteristic curve of the wicking geotextile through experiments and demonstrated the superiority of the wicking geotextile in soil drainage. Guo et al. [20] designed different geotextile scenarios, including wicking geotextiles, non-wicking geotextiles, and no geotextile, in different subgrades and compared the drainage and deformation effects of subgrades under cyclic loading. They found that a core suction geotextile had the best drainage and deformation control effects. Bai et al. [21] studied the influence of parameters such as the number and spacing of fibers on the drainage performance of wicking geotextiles and found that a geotextile with a single core suction fiber with a fiber yarn spacing of 3 mm had the best effect, reducing the soil water content by 11.31% from the initial value of 18%. Zornberg et al. [22] combined field and laboratory tests and summarized the numerous benefits of the new wicking geotextile in water absorption and drainage. Through field model tests, Lin et al. [16] further found that the water absorption and drainage effect of a new wicking geotextile was twice that of the initial design when combined with a plant system (i.e., planting vegetation at the exposed end of the wicking geotextile). Guo et al. [23] learned in soil column tests that the effective drainage range of a wicking geotextile in granular base layers was 180~250 mm, and the water content of the granular material could be reduced to 0.6% below the optimal water content. Through numerical simulations, Lin et al. [24] reported that adding wicking geotextiles to the subgrade reduced the moisture content by 2.2%.

The successful application of the new type of drainage system using wicking geotextiles has been further confirmed in some practical engineering projects. For example, Zhang et al. [25] laid two layers of new wicking geotextiles in an 18.3 m long section of the Dalton Highway in the United States, where the pumping phenomenon was the most severe, and found that the wicking geotextile could effectively eliminate the pumping phenomenon. Lin et al. [26] analyzed the long-term performance of new wicking geotextile based on five years of temperature and humidity data from the Dalton Highway test section and found that the wicking geotextile still worked effectively after five years. Using wicking geotextiles, Delgado [27] solved the problem of pavement life deterioration caused by differential settlement using the wicking geotextile.

It can be seen that current research mainly focuses on the drainage effect of geotextiles in coarse-grained soils, while its effectiveness in fine-grained soils requires further study. Moreover, geotextiles have enormous potential for solving roadbed wetting problems, and the study of water migration in fine-grained soils under the action of geotextiles has great research value and application significance. Therefore, in this paper, we explore the humidity control effect of a new wicking geotextile in fine-grained soil subgrades, using typical silty clay fillers. Based on theoretical and experimental results, the hydraulic properties of the wicking geotextile are quantified, and the absorption and drainage mechanisms are revealed. A subgrade water migration model considering the wicking geotextile is established, and the spatial distribution of subgrade water content before and after the installation of the wicking geotextile is obtained. The improvement in the subgrade support state before and after laying wicking geotextiles is compared.

## 2. Materials and Methods

### 2.1. Soil Parameters

The soil sample used in this study was a subgrade filler taken from the Ji-Qing Highway at a depth of 3 m in Jinan, China. A series of laboratory tests were conducted in accordance with the “Code for Geotechnical Testing of Highways” (JTG 3430-2020) [28] to obtain the basic physical properties of the soil samples, as shown in Table 1. The silty clay used in the test had a silt content of 85.92% and clay content of 11.08%. The liquid limit and plastic limit of the silty clay were 31.19% and 19.98%, respectively, with a plasticity index IP of 11.21. The maximum dry density was 1.91 g/cm^3^, and the optimal water content was 12.0%.

### 2.2. Wicking Geotextile Parameters

The new type of wicking geotextile is made by cross-weaving longitudinal polyester material and transverse green water-absorbing and draining yarn, as shown in Figure 1a. Yang et al. [29] found that the absorbing fibers provided capillary channels for water discharge and changed the matric potential in the soil through its capillary channels, and the difference in suction between inside and outside the soil provided the driving force for water discharge. The scanning electron microscopy experiment was carried out on the new wicking geotextile treated by liquid nitrogen-freezing the transverse water-absorbing and draining yarn. It was found that it was composed of bundles of fibers with approximately circular cross-sections, as shown in Figure 1b. Yao et al. [30] proposed a formula for calculating the equivalent diameter of inter-fiber pores in the textile yarns as,
(1)$$d_{eyf}=d_{f}\left[\varepsilon_{yf}/\left(1-\varepsilon_{yf}\right)\right]$$
where $d_{f}$ is the fiber diameter, equal to 28.775 μm; $\varepsilon_{yf}$ is the porosity of inter-fiber pores, equal to 0.236. According to Equation (1), the equivalent diameter of the pores between fibers is 8.88 μm. The contact angle tests on the water-absorbing and draining yarn showed a contact angle of 62.04°, indicating that it is a hydrophilic material with strong water-absorbing ability.

According to “Specification for test and measurement of geosynthetics” (SL 235-2012) [31], we performed a vertical permeability coefficient test and horizontal permeability coefficient test. The vertical and horizontal permeability coefficients of wicking geotextiles were measured, which were 1.3 × 10^−2^ cm/s and 2.2 cm/s, respectively. It can be seen that the horizontal permeability of wicking geotextiles is much greater than the vertical permeability. As a result, horizontal drainage is the main function.

### 2.3. Hydrophilic Properties of Wicking Geotextiles

The contact angle is an important indicator of the wetting ability of a liquid on a solid. When the contact angle θ < 90°, it indicates that the surface of the solid material is easily wetted and is a hydrophilic material. In this study, we used the dynamic capillary method to measure the contact angle of the wicking geotextile. The Washburn equation proposed by Lucas–Washburn, which describes the capillary flow of porous materials, states that the square of the rise height of the liquid is proportional to time, as shown in Equation (2),
(2)$$H^{2}=\frac{ra\cos\theta }{2\varepsilon }t$$
where H is the height of liquid diffusion in the capillary (m); ε is the viscosity of water, equal to 1.139 × 10^−3^ Pa·s; a is the interfacial tension between liquid and gas, equal to 0.07346 N/m; r is the equivalent radius of the pore between the fibers, equal to 4.44 × 10^−6^ μm; θ is the contact angle between the solid and the liquid; and t is time.

After testing the vertical capillary rise of the water, we plotted the relationship between the square of the liquid capillary rise height $H^{2}$ and time t, as shown in Figure 2. The average slope of the two fitting curves was 6.714 × 10^−5^. Using the Washburn equation, the contact angle was calculated to be 62.04°, indicating that the new wicking geotextile has good wettability and is a hydrophilic material.

### 2.4. Analysis of Water Content Characteristics of Silty Clay and Geotextiles

Through the pressure plate test, we obtained the volumetric water content of the silty clay and wicking geotextile at different matric suction levels. Based on the Van Genuchten model, the soil–water characteristic curves of the silty clay and wicking geotextile were established, as shown in Figure 3, and expressed by Equations (3) and (4).
(3)$$\theta_{1}=0.194+\frac{0.246}{{\left[1+{\left(0.17h\right)}^{4.013}\right]}^{0.751}}$$
(4)$$\theta_{2}=0.172+\frac{0.162}{{\left[1+{\left(0.012h\right)}^{2.202}\right]}^{0.546}}$$

Compared with the soil–water characteristic curve of the silty clay, the wicking geotextile had a lower suction value and worse water-holding capacity. The air entry values of the two materials were 20 kPa and 3 kPa, respectively. In addition, the linear change segment of the matric suction for the wicking geotextile was 3~15 kPa. When the matric suction reached 15 kPa, the volumetric water content rapidly decreased to 22.9%, while the volumetric water content of the silty clay was 32.8%. Such differences ensure that the wicking geotextile can continuously drain water from the soil.

### 2.5. Constitutive Equations of Numerical Model

According to the law of conservation of mass (5) and the generalized Darcy’s law (6), the Richard Equation (7) describing water flow in soil can be obtained. This equation is applicable to water flow in both saturated and unsaturated soils and can be used to investigate water flow under the combined effects of gravity potential, pressure potential, and matric suction.

Mass conservation:(5)$$-(\frac{\partial q_{x}}{\partial x}+\frac{\partial q_{y}}{\partial y}+\frac{\partial q_{z}}{\partial z})\rho_{l}=\frac{\partial \rho \theta }{\partial t}$$
where $q_{x}$ is the flux of water in the direction of x; $q_{y}$ is the flux of water in the direction of y; $q_{z}$ is the flux of water in the direction of z; $\rho_{l}$ is the density of liquid water; and θ is volumetric water content.

Generalized Darcy’s Law:(6)$$\left\{\begin{array}{l} q_{x}=-k_{x}\frac{\partial H}{\partial x}\\ q_{y}=-k_{y}\frac{\partial H}{\partial y}\\ q_{z}=-k_{z}\frac{\partial H}{\partial z} \end{array}\right.$$
where $k_{x}$ is the permeability coefficient in the direction of x; $k_{y}$ is the permeability coefficient in the direction of y; $k_{z}$ is the permeability coefficient in the direction of z; and H is the total water head.

Substituting Equation (6) into Equation (5) yields Richard’s equation as:(7)$$\rho_{l}\left[\frac{\partial }{\partial x}\left(k_{x}\frac{\partial H}{\partial x}\right)+\frac{\partial }{\partial y}\left(k_{y}\frac{\partial H}{\partial y}\right)+\frac{\partial }{\partial z}\left(k_{z}\frac{\partial H}{\partial z}\right)\right]=\frac{\partial \left(\rho \theta \right)}{\partial t}$$

Water storage capacity S is defined as the volume of water stored (or released) per unit volume of porous medium when the pore water pressure rises (or falls) by one unit:(8)$$S=\rho g\left(\left(1-n\right)\lambda +n\beta \right)$$
where λ is the compressibility of the soil skeleton; β is the compressibility of the fluid; and n is porosity.

Substituting Equation (8) into Equation (7), the Richard equation is extended to the saturated–unsaturated seepage equation with the variable being the pore water pressure p as,
(9)$$\rho_{l}\left(\frac{C}{\rho g}+S_{e}S\right)\frac{\partial p}{\partial t}-\nabla \rho \left[\frac{K_{s}k_{r}}{\rho g}\left(\nabla p+\rho g\nabla H\right)\right]=0$$
where ∇ is the gradient operator; H is the height difference; C is the specific storage coefficient; and $S_{e}$ is saturation. When the saturation is 1 (fully saturated), the specific storage coefficient is 0, and the equation becomes the differential equation for saturated flow.

### 2.6. Numerical Analysis and Feasibility of Water Flow Model

A numerical model for water flow considering the wicking geotextile was established based on the Richard equation in COMSOL. The soil parameters include saturated volumetric water content *θ*s = 33.4%, residual volumetric water content *θ*r = 17.2%, Van Genuchten model parameters α = 0.012, m = 0.546, n = 2.202, and permeability coefficient 5.5 × 10^−6^ cm/s. The wicking geotextile has the following parameters: saturated volumetric water content *θ*s = 44%, residual volumetric water content *θ*r = 19.4%, Van Genuchten model parameters α = 0.170, m = 0.751, n = 4.013, vertical permeability coefficient 1.3 × 10^−2^ cm/s, and horizontal permeability coefficient 2.2 cm/s, as shown in Table 2.

To validate the water flow model, we designed a chamber for silty clay with wicking geotextile laboratory tests. The size of the test chamber was 30 cm × 30 cm × 30 cm, as shown in Figure 4a. In order to realize the continuous monitoring of soil moisture content, the EC-5 moisture sensor was used in this experiment. A rectangular gap for the wicking geotextile was set at 10 cm from the surface of the soil on the front side. On the back side of the chamber, circular holes were drilled at distances of 3 cm, 8 cm, 12 cm, and 17 cm from the soil surface, corresponding to MS1, MS2, MS3, and MS4, respectively, where humidity sensors were placed. The specific positions of the geotextile and the humidity sensors are shown in Figure 4b. The optimal water content was used for layered compaction, and the compaction degree was controlled at 96%. After the entire test chamber was prepared, it was saturated with water, sealed, and the wicking geotextile was horizontally unfolded, while simultaneously recording the values of the humidity sensors in the chamber.

The entire model test took 51 days. After water was added, the positions 3cm and 8cm away from the soil surface (sensors MS1 and MS2) quickly reached saturation, and the volume water content increased to 38.1% and 37.8%, respectively. However, saturation was slightly delayed at positions 12 cm and 17 cm from the soil surface (sensors MS3 and MS4), and the volumetric water content increased to 37.2% and 34.4%, respectively. This phenomenon was caused by the interception of some upper infiltration water by the geotextile, which reduced the infiltration rate of water. As the test progressed, the volume water content at positions 3 cm and 17 cm from the soil surface (MS1 and MS4) decreased significantly, while the volume water content at positions 9 cm and 12 cm (MS2 and MS3) remained unchanged or slightly fluctuated, as shown in Figure 5. At the end of the test, the volume water content at positions 3 cm, 8 cm, 12 cm, and 17 cm away from the soil surface (MS1, MS2, MS3, and MS4) decreased by 3.62%, 2.54%, 1.64%, and 2.80%, respectively, compared with the saturated state. The decrease in volumetric water content was more significant at positions 3 cm and 17 cm away from the soil surface (MS1 and MS4) than at positions close to the wicking geotextile. This indicates that while the geotextile drains water from the adjacent soil to the external environment, it also continuously receives upper gravity water and lower capillary water.

The same size of two-dimensional soil column model was built in COMSOL based on the laboratory model described above, and the model was meshed and transiently solved using triangular elements. Four specific time points were selected to compare and analyze the simulation results with the experimental results (13 d, 26 d, 39 d, and 51 d), as shown in Figure 6. Generally, the soil column volumetric water content showed an increasing and then decreasing pattern in the depth direction. The simulation results were in agreement with the experimental measurements, and the volumetric water content reached its peak value near the geotextile. Table 3 shows the comparison of soil column volume water contents at different depths and time points. The maximum error between the simulated and measured volumetric water content was 3.6%, and the average error was 1.8%, which was within a reasonable range. This further demonstrates the feasibility of the water flow model based on the COMSOL software, version 5.6.

## 3. Numerical Results and Discussion

An unsaturated soil subgrade water flow model was selected for a typical two-way four-lane highway structure with a roadbed width of 28 m, a height of 4 m, and a slope of 1:1.5. The groundwater level in the Yellow River flood-prone area is usually 2~10 m below the ground surface. Four working conditions were selected in the analysis: a groundwater table of 2 m, 4 m, 6 m, and 8 m below the subgrade base. A layer of geotextile was laid at the bottom of the roadbed. The road structure was symmetric, and the model was built for the right side. A COMSOL numerical model was established as shown in Figure 7. The geotextile was divided by free quadrilateral mesh, and the rest was divided by mapped mesh. The total number of elements was 1,246,752. Four characteristic sections were selected for analysis: x = 0 m (centerline of the subgrade), x = 6 m (outer edge of the first lane), x = 9.75 m (outer edge of the second lane), and x = 13.25 m (outer edge of the hard road shoulder).

The initial mass water content of the subgrade was 12%, which translates to a volumetric water content of 22%. The initial suction can be calculated using Equation (3) as 220 kPa. As the model is for the right side of a symmetrical subgrade structure, the left boundary of the model was set as a symmetric boundary. It was assumed in this study that the pavement structure was not cracked and did not allow water flow; thus, the upper boundary of the model was set as a no-flow boundary. The relevant input parameters of the model are described in previous sections. The relevant parameters of the model are described in Table 2.

### 3.1. Matric Suction and Volumetric Water Content Distribution

When the groundwater level was 4 m from the bottom of the subgrade, based on the steady-state analysis using COMSOL, the distribution contours of the volumetric water content and matric suction inside the subgrade before and after the installation of the wicking geotextile were obtained, as shown in Figure 8 and Figure 9. Overall, the distribution of volumetric water content and matric suction at the same depth in the subgrade varied along the cross-sectional direction, with higher matric suction and lower volumetric water content further away from the subgrade centerline, and higher matric suction and lower volumetric water content with increasing distance from the groundwater level. Moreover, before the installation of the wicking geotextile, the volumetric water content of the subgrade reached 26.9~29.5%, much higher than the optimal value of 22%. After the installation of the wicking geotextile, there were apparent changes in both the matric suction and volumetric water content above and below it, with the increase in matric suction in the subgrade ranging from 68.9 kPa to 107.4 kPa and the volumetric water content above the geotextile decreasing to 17.9% to 21.9%, indicating that wicking geotextiles play a significant role in water insulation.

### 3.2. Matric Suction Distribution

Figure 10 shows the distribution of subgrade matric suction along the depth of the four specific sections with a groundwater table of 4m. Without the geotextile, the matric suction at the centerline of the subgrade increased linearly with the depth from the groundwater table, from 40 kPa to 71.1 kPa. At the same time, the matric suction in the direction of the cross-section at the same depth decreased with the increase in distance from the subgrade centerline, and the matric suction at the top surface of the subgrade increased from 71.1 kPa at the centerline to 94.6 kPa at the shoulder. Because the volumetric water content of subgrade is controlled by matric suction, without geotextiles, the volumetric water content in the entire depth of the subgrade exceeded the initial optimal volumetric water content of 22%.

After the installation of the wicking geotextile at the bottom of the subgrade, the volumetric water content at different depths changed significantly. This is mainly due to the presence of the wicking geotextile, which significantly increased the matric suction within the subgrade. The maximum increase in matric suction for each characteristic section is shown in Figure 11. At distances of 0 m, 6 m, 9.75 m, and 13.25 m from the centerline of the subgrade, the maximum increases in matric suction were 62.3 kPa, 69.1 kPa, 82.5 kPa, and 107.4 kPa, respectively. The matric suction in the entire subgrade increased from 71.1~94.6 kPa to 134.3~203.9 kPa, indicating that the wicking geotextile has a strong ability to block the invasion of capillary water from the lower layer. Lin et al. [32] found that the soil close to the wicking geotextile was further dried as time passed by and the average suction of the overlying soil reached 252.0 kPa after 1 month, indicating the wicking geotextile has the capability to impede water from passing through in the cross-plane direction. In addition, the volumetric water content of the subgrade within the range of 6~13.25 m from the centerline already reduced to below the initial optimal water content of 22%. This is because a certain suction gradient is formed laterally along the geotextile, and water inside the subgrade can migrate to the subgrade slope through the fiber channels and finally evaporate into the surrounding atmosphere. This further indicates that the wicking geotextile has the ability to absorb and drain water and can effectively absorb and drain weakly bound water in unsaturated soil.

### 3.3. Volumetric Water Content Distribution

For the condition of groundwater level between approximately 2 and 8 m, the average humidity of the subgrade area with and without geotextile is shown in Figure 12. As the groundwater level decreased, the average volumetric water content of the subgrade area decreased approximately linearly, but it always remained higher than the optimal volumetric water content. For example, the volumetric water content of the outside edge of the first lane at the groundwater levels of 2 m, 4 m, 6 m, and 8m was 30.9%, 29.1%, 27.5%, and 26.1%, respectively, and the increase in water content led to a decrease in the modulus of the subgrade. Although the average volume water content of the roadbed at a groundwater level of 2 m was still higher than the optimal water content, it decreased rapidly as the groundwater level dropped from 2 m to 4 m. Then, as the groundwater level fell from 4m to 6~8 m, the volumetric water content remained almost constant. For example, the volumetric water content of the outside edge of the second lane at the groundwater level of 2 m, 4 m, 6 m, and 8 m was 24.9%, 21.6%, 21.5%, and 21.4%, respectively. The results show that the drainage effect of the wicking geotextile is related to the depth of the groundwater level, and under the condition of high groundwater level, multiple layers of geotextile may be required.

Figure 13 shows the distribution of the volumetric water content of the subgrade along the depth over four cross-sections at a groundwater level of 4 m. Without the geotextile, the volumetric water content at the centerline of the subgrade decreased linearly with increasing depth from the groundwater level, from 31.0% to 26.4%. This decrease was closely related to the linear increase in matric suction of the subgrade with respect to the distance from the groundwater level. Similarly, at the same depth, the volumetric water content across the subgrade decreased with increasing distance from the subgrade centerline, from 28.9% at the centerline to 26.9% at the shoulder of the road. Overall, without the geotextile, the volumetric water content throughout the entire depth of the subgrade exceeded the initial optimal value of 22%. 

After the installation of the wicking geotextile at the bottom of the roadbed, there was a notable change in the distribution of volumetric water content along the depth of the subgrade. The water content in the roadbed above the wicking geotextile decreased significantly, with maximum reductions of 7.6%, 8.0%, 8.8%, and 9.6% observed at the four selected cross-sections, as shown in Figure 14. The average water content above the wicking fabric was 4.2% lower than the initial optimal value of 22%. Lin et al. [16] used numerical models to evaluate the performance of pavement structures with wicking fabric. The simulation results showed that the soil–geotextile system was able to reduce the water content of the base course by 2.2% (volumetric water content 4.8%) from the optimum value.

### 3.4. Distribution of Saturation

Saturation is commonly used to describe the degree to which the pores in a soil are filled with water. Based on the conversion relationship of soil three-phase properties, saturation can be written as
(10)$$S_{r}=\frac{G_{s}\omega_{v}}{\gamma_{d}e}$$
where $S_{r}$ is saturation; $G_{s}$ is the specific gravity of soil particles; $\omega_{v}$ is the volumetric water content; $\gamma_{d}$ is the maximum dry density; and e is the void ratio.

Figure 15 shows the distribution of saturation of the subgrade along the depth at the four selected cross-sections. Without the geotextile, the saturation at the centerline of the subgrade decreased linearly with increasing depth from the groundwater level, reducing from 85.9% to 79.0%. Similarly, at the same depth, the saturation decreased with increasing distance from the subgrade centerline, reducing from 79.0% at the subgrade centerline to 74.3% at the shoulder. Overall, without the geotextile, the saturation remained relatively high throughout the depth of the subgrade.

After the installation of the geotextile at the bottom of the roadbed, the distribution of saturation along the depth underwent significant changes, and there was a sharp transition in saturation at the interface of the geotextile. The saturation in the roadbed area above the geotextile decreased significantly, with the maximum reduction in saturation at the four cross-sections reaching 18.3%, 19.7%, 21.4%, and 23.0%, as shown in Figure 16.

### 3.5. The Dynamic Resilient Modulus of the Subgrade

The dynamic resilient modulus of the subgrade is an important parameter for the design of pavement structures. The “Code for Geotechnical Testing of Highways” (JTG 3430-2020) gives an empirical equation for the dynamic resilient modulus of the subgrade as:(11)$$M_{R}=k_{1}p_{a}{\left(\frac{\theta }{p_{a}}\right)}^{k_{2}}{\left(\frac{\tau_{oct}}{p_{a}}+1\right)}^{k_{3}}k_{3}$$
where $M_{R}$ is the dynamic resilient modulus; $p_{a}$ is atmospheric pressure, equal to 101 kPa; θ is the body stress; $\tau_{oct}$ is the octahedral shear stress; and $k_{1}$, $k_{2}$, and $k_{3}$ are model parameters.

For fine-grained soils, $k_{1}$, $k_{2}$, and $k_{3}$ are defined as:(12)$$\left\{\begin{array}{l} k_{1}=-0.096\omega +0.3929\rho_{d}+0.0142I_{p}+0.0109P_{0.075}+1.01\\ k_{2}=-0.0005\omega -0.0069I_{p}-0.0026P_{0.075}+0.6984\\ k_{3}=-0.2180\omega -3.0253\rho_{d}-0.0323I_{p}+7.1474 \end{array}\right.$$
where ω is mass water content of the specimen, $\rho_{d}$ is the dry density of the specimen, $I_{p}$ is the plasticity index of the specimen, and $P_{0.075}$ is the fine particle content of the specimen.

For the silty clay filler used in this study, the dynamic resilient modulus at the optimal mass water content of 12% was 133 MPa. The dynamic resilient modulus of the subgrade under different groundwater levels can be calculated using Equation (11), as shown in Figure 17. When the groundwater level was 2 m, the dynamic resilient modulus of the subgrade for the inner before and after the installation of the geotextile was 86.2 MPa and 112.5 MPa, respectively, indicating an increase of 30.5%. When the groundwater level dropped to 4~8 m, the dynamic resilient modulus of the subgrade before and after the installation of the geotextile ranged from 94~110 MPa to 135.3~136.8 MPa, respectively, representing an increase of 24.4~43.9%. Lin et al. [16] validated the ability of a soil–geotextile system to reduce the water content of the base course, and the corresponding resilient modulus could be increased by 2–3 times from the optimum value. This indicates that a geotextile, by controlling the water content in the roadbed, can effectively improve the dynamic resilient modulus of the subgrade and extend the service life of the road.

## 4. Conclusions

This study demonstrated the working mechanism of a wicking geotextile with lateral drainage abilities. Based on a physical model of wicking geotextile-reinforced silty clay, we investigated the development of soil moisture content, matric suction, and the dynamic round modulus with or without the wicking geotextile under different water tables. The major conclusions are summarized as follows:(1)The new type of wicking geotextile has an equivalent pore diameter of 8.88 μm, contact angle of 62.04°, air entry pressure of only 3 kPa, and vertical and horizontal permeability coefficients of 1.3 × 10^−2^ cm/s and 2.2 cm/s, respectively. This indicates that the new wicking geotextile has strong hydrophilicity, low water-holding capacity, and high horizontal permeability.(2)A numerical model considering water flow with the presence of the wicking geotextile was established based on Richard’s equation. By comparing the volumetric water content of each layer of the soil column from the finite element software COMSOL and laboratory experiments, it was found that the numerical model and experimental results exhibited a similar trend in volumetric water content. The maximum error was found to be 3.6%, and the average error was 1.8%, demonstrating the feasibility of the water flow model.(3)The wicking geotextile is effective for controlling the water content of the subgrade. After the installation of the geotextile, there was a noticeable abrupt change in the volumetric water content above and below the geotextile. At a groundwater level of 4m, the volumetric water content above the geotextile decreased significantly by 7.6% to 9.6%, and the saturation decreased significantly by 18.3% to 23.0%. The matric suction increased by a large factor of 2~2.3 times, indicating that the wicking geotextile effectively acts as a barrier to water infiltration.(4)By controlling the internal water content, the wicking geotextile can effectively increase the dynamic resilient modulus of the subgrade. When the groundwater level was 4 m or below, the dynamic resilient modulus of the subgrade increased from 94.2~111 MPa without the geotextile to 135.3~136.8 MPa with the geotextile, representing an improvement of 23.2~43.6%.

In summary, the typical silty clay subgrade in Shandong Province was selected in this paper to study the moisture-controlling effect of wicking geotextiles under different groundwater levels, providing an effective solution to the wetting problem of the unsaturated subgrade of fine-grained soil.

## Figures and Tables

**Figure 1 materials-17-00390-f001:**
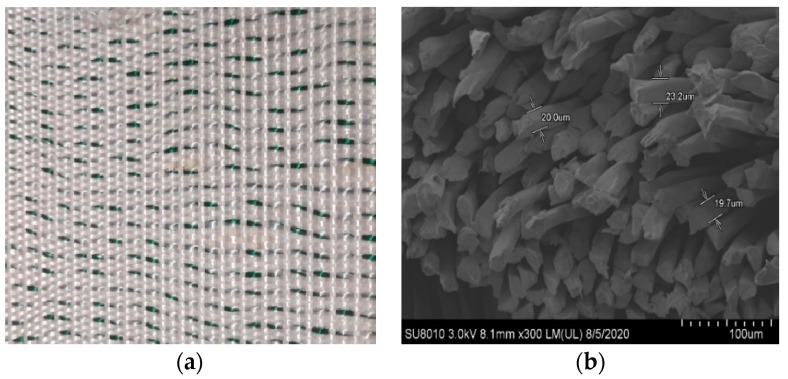
The morphology of wicking geotextile: (**a**) physical example of wicking geotextile and (**b**) microstructure of wicking and drainage yarn.

**Figure 2 materials-17-00390-f002:**
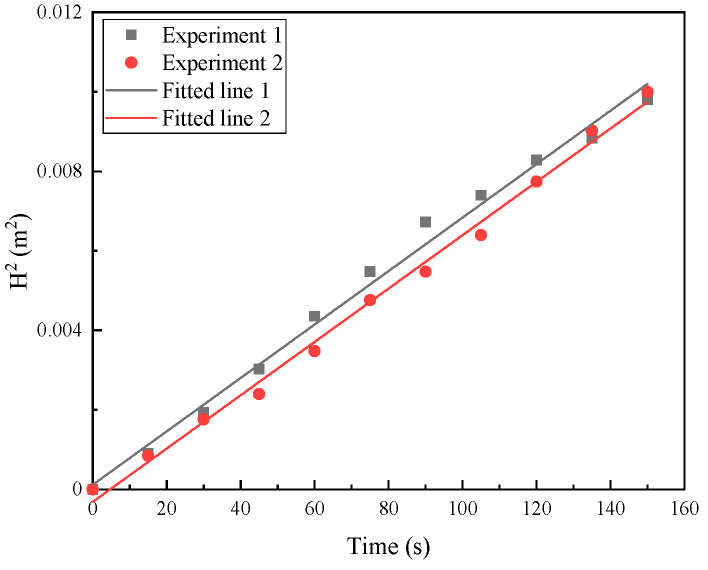
Relation of H^2^ with time.

**Figure 3 materials-17-00390-f003:**
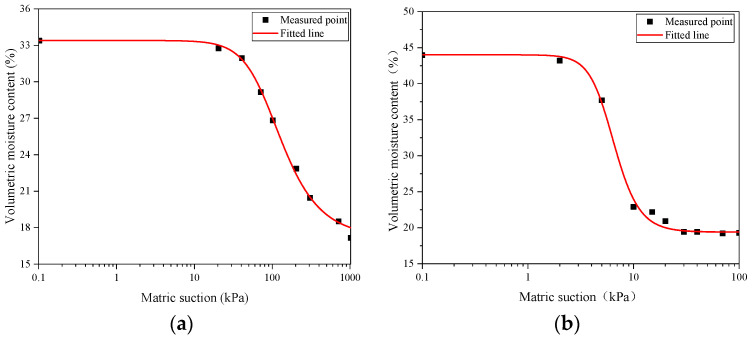
Water absorption characteristic curve of silty clay and wicking geotextile: (**a**) silty clay and (**b**) wicking geotextile.

**Figure 4 materials-17-00390-f004:**
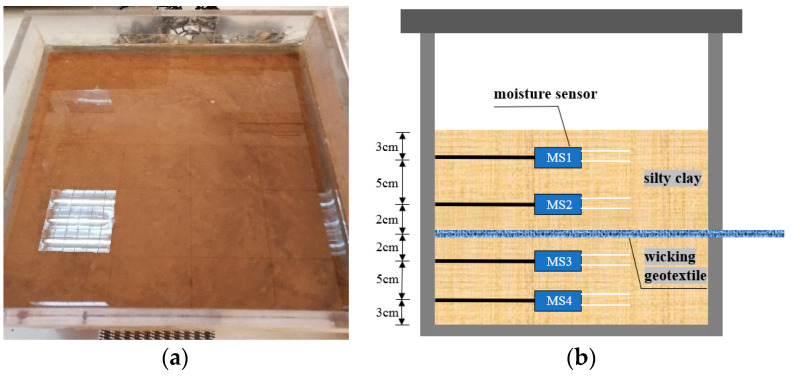
Water flow test chamber: (**a**) specimen saturation process and (**b**) schematic diagram of the test chamber.

**Figure 5 materials-17-00390-f005:**
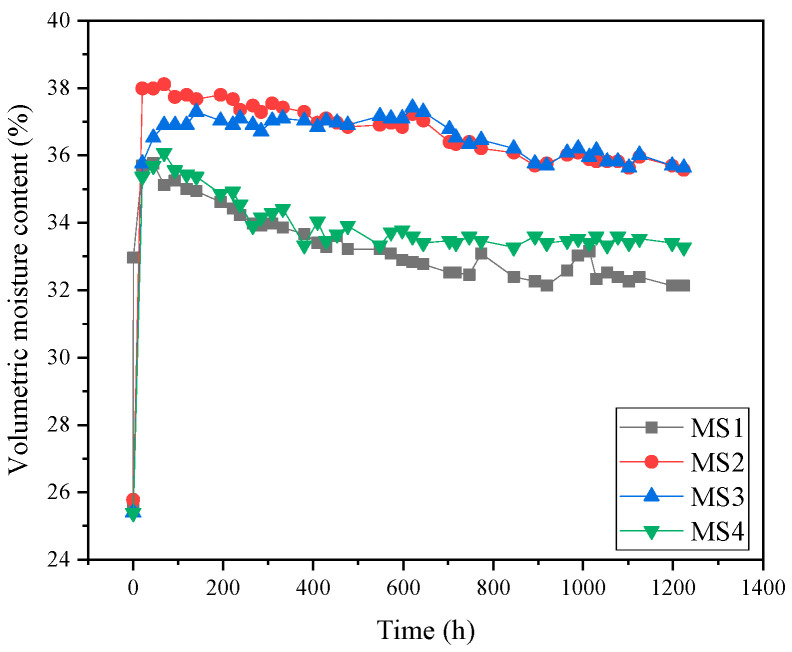
Volumetric water content as a function of time.

**Figure 6 materials-17-00390-f006:**
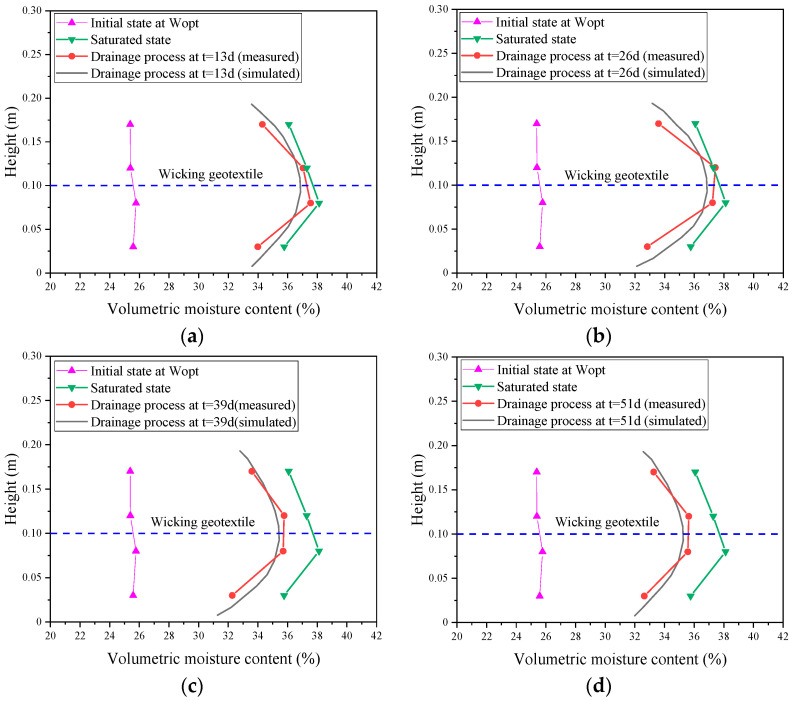
Comparison of volumetric water contents of subgrade at different time points: (**a**) t = 13 d, (**b**) t = 26 d, (**c**) t = 39 d, and (**d**) t = 51 d.

**Figure 7 materials-17-00390-f007:**
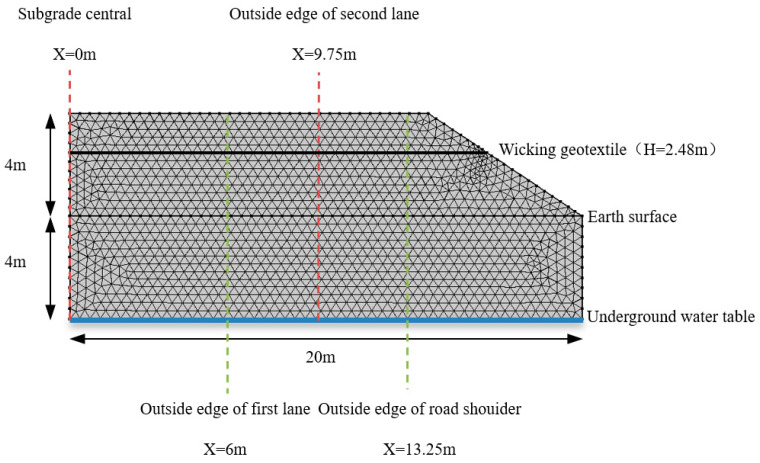
Numerical model for water flow in the subgrade.

**Figure 8 materials-17-00390-f008:**
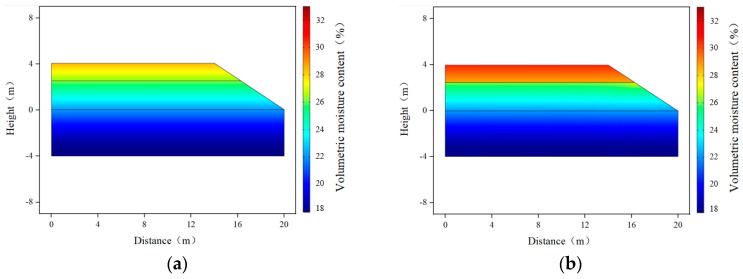
Distribution contour of subgrade volumetric water content: (**a**) without wicking geotextile and (**b**) with wicking geotextile.

**Figure 9 materials-17-00390-f009:**
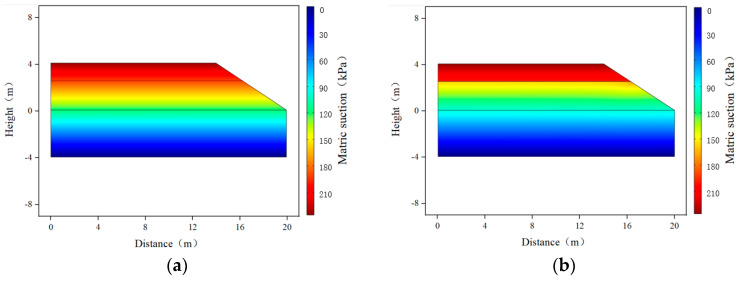
Distribution contour of subgrade matric suction: (**a**) without wicking geotextile and (**b**) with wicking geotextile.

**Figure 10 materials-17-00390-f010:**
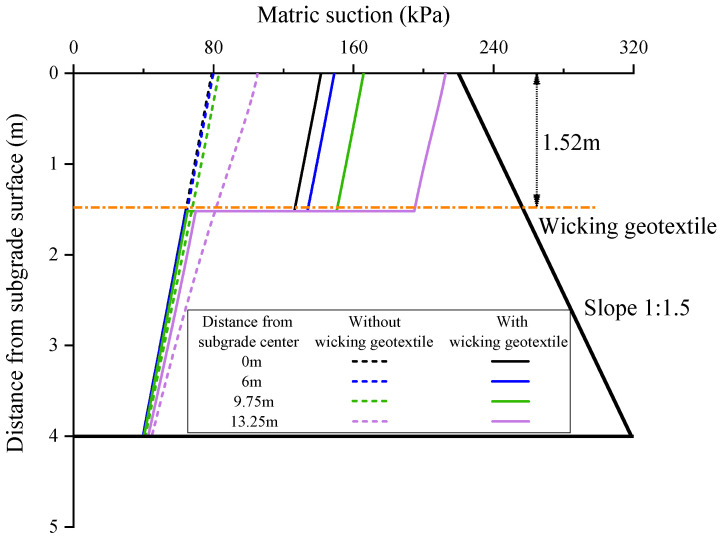
Vertical distribution of matric suction of the subgrade.

**Figure 11 materials-17-00390-f011:**
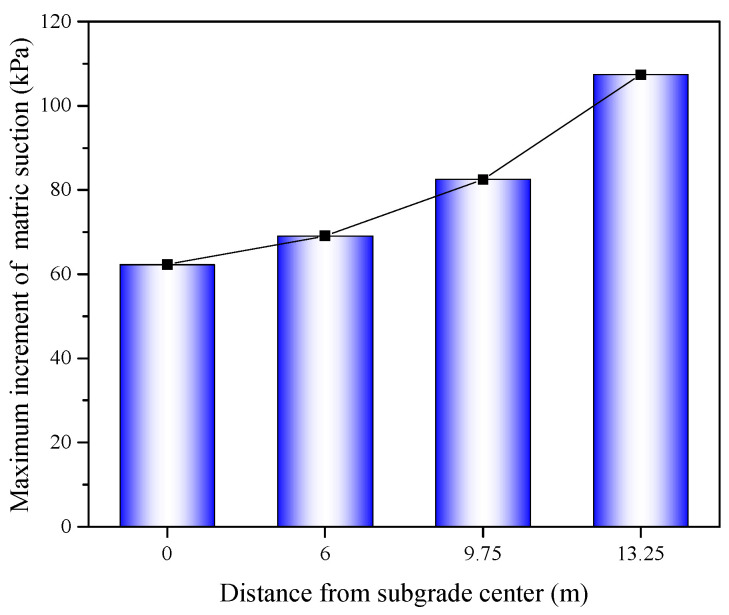
Maximum reduction in matric suction at the different distances from the subgrade centerline.

**Figure 12 materials-17-00390-f012:**
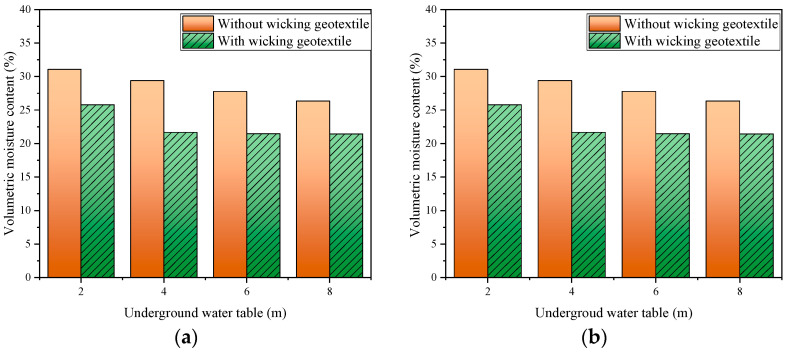
Volumetric water content of the subgrade at different groundwater levels: (**a**) 0 m from the subgrade centerline, (**b**) 6 m from the subgrade centerline, (**c**) 9.75 m from the subgrade centerline, and (**d**) 13.25 m from the subgrade centerline.

**Figure 13 materials-17-00390-f013:**
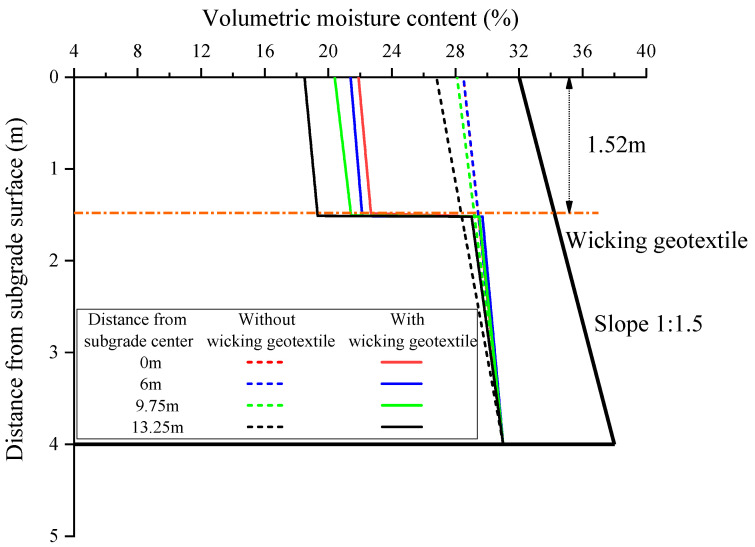
Distribution of volumetric water content of the subgrade in four different cross-sections.

**Figure 14 materials-17-00390-f014:**
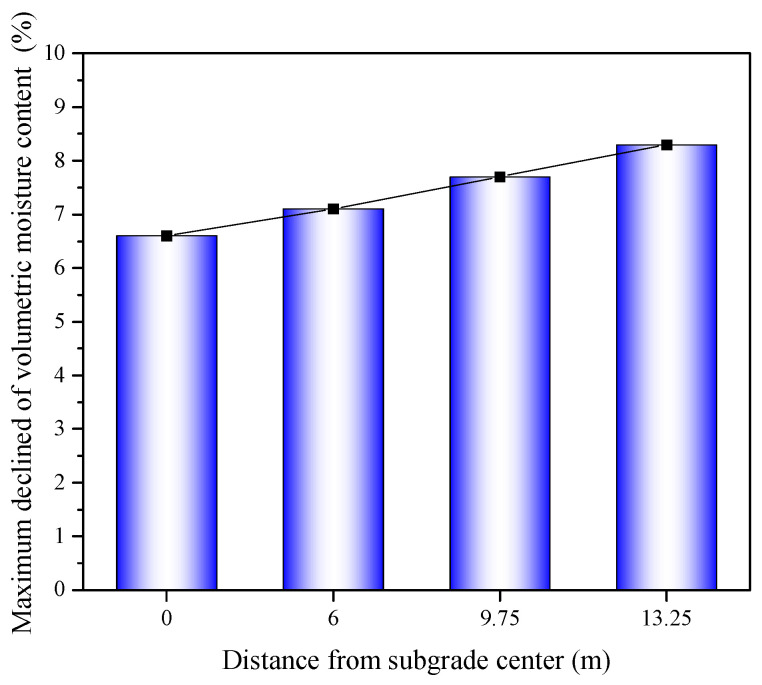
Maximum reduction in volumetric water content at four different cross-sections.

**Figure 15 materials-17-00390-f015:**
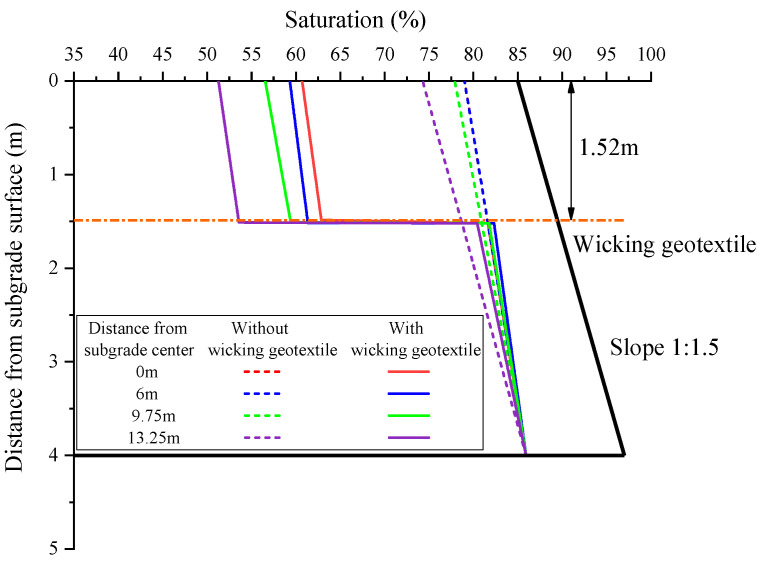
Distribution of saturation of the subgrade at four different cross-sections.

**Figure 16 materials-17-00390-f016:**
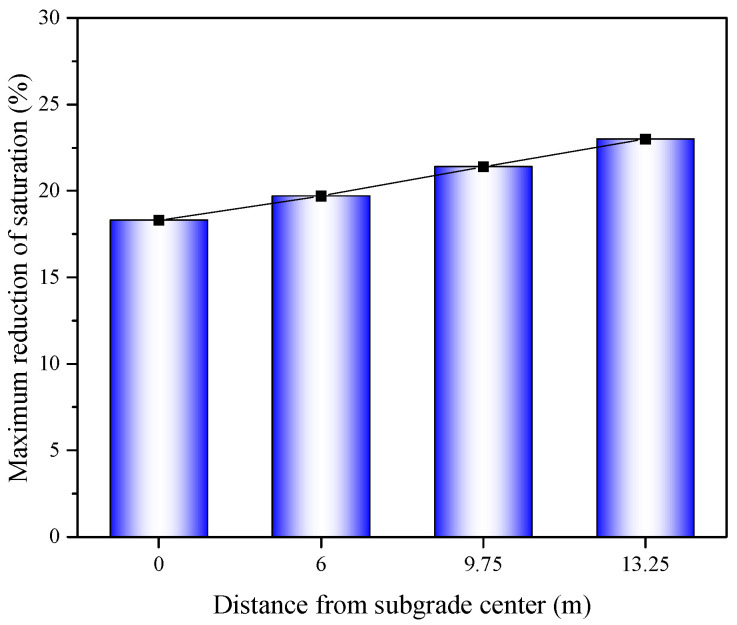
Maximum reduction in saturation of the subgrade at four different cross-sections.

**Figure 17 materials-17-00390-f017:**
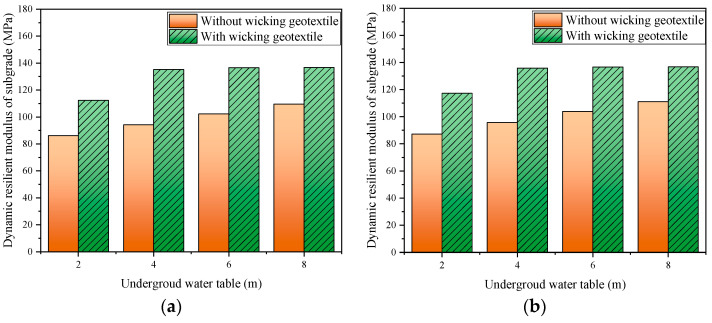
Dynamic resilient modulus of the subgrade with or without geotextiles corresponding to different groundwater levels: (**a**) 9.75 m from the subgrade centerline and (**b**) 13.25 m from the subgrade centerline.

**Table 1 materials-17-00390-t001:** Physical properties of the soil sample.

Maximum Dry Density/(g/cm^3^)	Optimal Water Content/%	Liquid Limit/%	Plastic Limit/%	Plasticity Index	Particle Size Distribution/%		
					<5 μm	5~75 μm	75~250 μm
1.91	12	31.19	19.98	11.21	16.9	80.1	3.0

**Table 2 materials-17-00390-t002:** Parameters of the VG numerical model.

Parameter		Value	
		Soil	Wicking Geotextile
saturated volumetric water content		33.4%	44%
residual volumetric water content		17.2%	19.4%
VG model parameters	α	0.012	0.170
	m	0.546	0.751
	n	2.202	4.013
vertical permeability coefficient		5.5 × 10^−6^ cm/s	1.3 × 10^−2^ cm/s
horizontal permeability coefficient		5.5 × 10^−6^ cm/s	2.2 cm/s

**Table 3 materials-17-00390-t003:** Volumetric water contents of soil at different time points.

Time/d	Measurement Point	Measured Result/%	Simulated Result/%	Error/%
13	MS1	34.3	35.1	2.3
	MS2	37.0	36.6	1.1
	MS3	37.5	36.7	2.1
	MS4	34.0	34.3	0.9
26	MS1	33.6	34.8	3.6
	MS2	37.4	36.7	1.9
	MS3	37.2	36.6	1.6
	MS4	32.8	33.9	3.6
39	MS1	33.6	33.9	0.9
	MS2	35.8	35.2	1.7
	MS3	35.7	35.3	1.1
	MS4	32.3	33.2	2.8
51	MS1	33.3	33.7	1.2
	MS2	35.6	35.1	1.4
	MS3	35.6	35.2	1.1
	MS4	32.6	33.2	1.8

## Data Availability

Data are contained within the article.

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
