# Peer review of "A Numerical Simulation of Moisture Reduction in Fine Soil Subgrade with Wicking Geotextiles"

_materials, 2024, doi:10.3390/ma17020390_

Round 1
Reviewer 1 Report
Comments and Suggestions for Authors
1. What is the main question addressed by the research?
The research in the paper consists of analyzing the drainage capacity of a wicking geotextile using numerical modelling supported by laboratory experiments.
2. Do you consider the topic original or relevant in the field? Does it address a specific gap in the field?
I consider the topic relevant. The paper is not innovative, but some innovation could be found related to the fact that the paper investigates the moisture reduction in fine soils of the pavement subgrade due to the new wicking geotextile. This is the specific gap covered by the paper in using geotextiles in pavement subgrade to control moisture content.
3. What does it add to the subject area compared with other published material?
The objectives and methodology.
4. What specific improvements should the authors consider regarding the methodology? What further controls should be considered?
In my opinion, the authors should clarify the durability of the geotextile, maintaining the capacity to control the moisture content of the subgrade.
5. Are the conclusions consistent with the evidence and arguments presented, and do they address the main question posed?
The main findings are well described in the manuscript and consistent with the results. However, the section of conclusions provides the main findings of the study. A brief description of the objectives and methodology of the study will be helpful for the reader in this final section. The section should be dedicated to the main conclusions. Further studies on the continuity are also necessary.
6. Are the references appropriate?
Yes. I recommend confirming the references format (e.g., the year of the reference in “bold”).
7. Please include any additional comments on the tables and figures.
Line 104 mentions Table 1, but I do not find the table in the manuscript. This table is fundamental to understanding the main properties of the soil used in the study. There is another Table 1 (Line 212). Table 1 should be included, and tables should be renumbered. Correct “Matric suction” (Matrix suction) in the horizontal axle of Figure 10 and the vertical axles of Figures 9 and 11.
Other comments:
a) Another main concern to the wicking geotextile is durability. Have the authors performed any study regarding this phenomenon? Some comments regarding this issue should be included in the manuscript.
b) What type of hydraulic experiment was performed by the authors (line 131)?
c) What moisture sensors were used in the experiment of Figure 4?
Reviewer 2 Report
Comments and Suggestions for Authors
The manuscript is well structured, and the subject fits into The Materials Journal. Some comments were listed for manuscript improvements.
Abstract - The main findings were described. However, it is important to show the Wicking geotextile applications (soil type, subgrade, capillary prevention, indication, etc.).
Page 1, lines 30-32: "Reports have shown that ... for expressways." - What were the references about the reports since just one reference was cited?
The introduction doesn't effectively bridge the gap between the research's main contribution and existing trends in the field.
Figures 6, 10 and 13: Please increase the font size.
Table 1 - Insert a legend of the VG model, which means VG, to make it more accessible for the readers.
The discussion section lacks information highlighting the study's relevance and findings. Maybe a comparative analysis with previous studies will make the section discussion stronger.
The study application must be described in more detail, which means soil type, local problems, subgrade water content, and places (soil) with a probability of the occurrence of what? What was the main application of your research? What about the recommendations and the limitations?
Comments on the Quality of English LanguageMinor editing of English language required
Round 2
Reviewer 1 Report
Comments and Suggestions for Authors
I acknowledge the authors for their efforts to answer the comments and improve the quality and scientific merit of the paper.